# Thermoregulation in Ectodermal Dysplasia: A Case Series

**DOI:** 10.3390/ijerph16224514

**Published:** 2019-11-15

**Authors:** Heather Massey, James House, Michael Tipton

**Affiliations:** Extreme Environments Laboratory, School of Sport, Health and Exercise Science, University of Portsmouth, Portsmouth PO1 2ER, UK

**Keywords:** hyperthermia, hereditary disorders, hypohidrosis

## Abstract

Ectodermal dysplasia (ED) is a rare genetic disorder occurring as a consequence of gene mutations that code for the ectoderm of the developing embryo and results in numerous disorders of varying severity. The lack of functioning sweat glands in those affected with ED leads to high infant mortality and frequent complaints of hyperthermia. Temperature control of two adolescents affected with ED was assessed by conducting heat and exercise exposures while monitoring insulated auditory canal (T_ac_) and skin temperatures, sweating rates, and skin blood flow. One participant was able to sweat and regulate his T_ac_ while a second participant could not regulate T_ac_ without a cooling intervention. The heterogeneous nature of ED, and these cases highlight the need for a case-by-case review of temperature control of individuals affected with ED. This will determine cooling strategies that would be of most benefit to the individual.

## 1. Introduction

Ectodermal dysplasia (ED) is a rare disorder resulting from mutations of genes coding for the ectoderm (primarily Ectodysplasin-A and Ectodysplasin-A receptor genes) and most commonly inherited as an X-linked recessive form, although autosomal (dominant and recessive) and spontaneous forms also exist [1]. The condition can be diagnosed either genetically (from a blood test or mouth swab) if there is a known family history, or clinically by the appearance of a triad of disorders: Few and often pointed teeth (hypodontia), fine and sparse hair (hypotrichosis), and diminished or absent sweat function (hypohidrosis), although there are many other conditions (including cleft lip and palate, eczema, xerophthalmia, xerostomia) that can co-exist and variably affect the patient [1]. X-linked hypohidrotic ectodermal dysplasias accounts for 70% of ED cases [2] and has an estimated prevalence of between 1.6 and 22 per 100,000 live births [3]. In the 1980s, the mortality rate was reported as 28% in the first three years of life [4]. Since then, further evidence suggests that this rate has been reduced to 2%, which is still greater than in healthy infants [5]. The improved prognosis over time is likely due to greater recognition of the condition [5].

Many patients affected with ED experience difficulties regulating their temperature and can develop hyperthermia as a result of reduced or absent sweating. Sweat production on the surface of the skin aids heat loss through evaporation. Consequently, a reduction or the absence of sweating may result in excess heat storage and the development of hyperthermia (a deep body temperature greater than 39 °C) and heat illness (including heat exhaustion or heat stroke) or death [4]. Unpublished data from the UK patient group of the Ectodermal Dysplasia Society (EDS) suggests that deep body temperature measurements of affected children are hotter than siblings without ED, when measured three times a day for a week. In that study, temperatures were recorded using infrared ear thermometers three times a day, for one week. More recently, a controlled laboratory-based study was performed [6] by exercising children with ED and matched controls (matched for sex and age) in 25 °C temperatures (40% relative humidity). The authors reported that the children with ED had deep body temperatures 1 °C higher after a maximum of 30 min of cycling exercise at a moderate to high intensity. Ninety minutes after the exercise, deep body temperatures of the ED group were still elevated above their starting temperatures whereas the control had cooled within 30 min of stopping. This suggests that children with ED have a rapid and possibly dangerous rise in temperature and may take longer to cool. In those affected by ED, cooling interventions could help to reduce the thermal strain experienced and increase the rate of cooling during recovery from exercise or heat exposure. Cooling interventions can increase heat loss through conductive, convective, or evaporative means. The effectiveness of the intervention may be dependent on the location of the cooling and whether a viable peripheral circulation can be maintained [7,8]. For instance, post-exercise ice vest use in ED patients resulted in greater rates of cooling in comparison to exercise recovery without cooling [4]. However, research in healthy volunteers exposed to high ambient temperatures showed whole body spraying and fanning and hand immersion in cold water had greater cooling power than air, liquid, or phase change cooling vests [7]. The question then arises whether or not a common approach to cooling can be adopted for all children with ED, or whether differences exist in the thermoregulatory capabilities of those with ED, and therefore cooling strategies should be tailored to these differing capabilities. 

## 2. Case History

Participant 1: Twelve-year old male (height 1.58 m, weight 53.6 kg) in the 92nd percentile for body weight for his age [9]. He has a clinical diagnosis of ED, based on the presentation of hypodontia, hypotrichosis, and assumption of hypohidrosis. Following advice from clinicians, at the time of diagnosis, his parents restricted his exposure to heat and participation in physical activity to prevent him overheating.

Participant 2: Seventeen-and-a-half-year-old male (height 1.67 m, weight 61.6 kg) trainee professional football player who ‘overheats during pre-season training and warm weather training camps’. He had a genetic diagnosis of X-linked ED; however, the gene mutation was not stated on the geneticist’s letter. He ‘managed’ by stopping exercise or acting only as a substitute. However, he was at the stage where he needed to show his footballing talent to progress to a professional contract, hence spending more time exercising at high intensities and in warm environmental conditions, for example, during pre-season training, early season, and end-of-season competition. 

## 3. Materials and Methods 

### 3.1. Experimental Overview

The testing was given a favorable ethical opinion by the University of Portsmouth Science Faculty Research Ethics Committee (SFEC 2014-088) and participants and parents, where appropriate, gave their informed consent and children gave assent to participate in the testing (in accordance with good clinical practice). 

Participants performed two bouts of intermittent treadmill exercise within a temperature-controlled laboratory (30 °C and 50% relative humidity). The duration and intensity of the bouts were dictated by the participant’s capability, with the intention to complete two bouts of 30 min of exercise. Exercise was halted at 30 min, at a T_ac_ of 39.5 °C, or at the participants request. The exercise bouts were separated by a passive recovery period. This varied in duration depending on the patient, and the final exercise session was followed by a cooling intervention, either immersion of the hands and/or feet in cold water (water temperature between 10 to 12 °C). This temperature was readily available from the tap and was used in previous research [8]. 

### 3.2. Measurement Techniques

The body mass of participants was recorded pre- and post-heat assessment. Before the exercise and heat assessment, participants were instrumented with an insulated auditory canal thermistor (T_ac_, Grant instruments, Cambridge, UK). For the assessment of deep body temperature, three lead Electrocardiogram (Fukuda Denshi, UK) to record heart rate were used, and sweat capsules on the right side of the upper chest and mid back (Q-Sweat, WR Medical Electronics Co, US) were used to determine the sweat rate. These locations have high sweat rates in healthy volunteers [10] and will likely be activated with the heat and exercise stimulus used. Laser Doppler sensors placed on the middle finger pad and ventral surface of the forearm (VM7-X Moor Instruments, UK) were used to measure skin blood flow, and the Laser Doppler probes were calibrated using a microsphere solution (Moors Instruments, UK), and biological zero was assessed by occluding flow to the forearm or digit. Data were corrected for the biological zero. In addition, skin heat flux sensors (FM-060-TH44033-F13, Concept Engineering, US) on the chest, mid bicep, mid-thigh, mid shin, mid forearm (next to the laser Doppler sensor), and finger pad of the index finger measured skin temperature and the direction of the thermal gradient. These data were logged at minute intervals using a power lab analogue to a digital recorder and chart software (AD Instruments). The mean skin temperature was calculated at minute intervals using four skin temperatures (from all sites except the forearm and finger) in accordance with Ramanathan [11]. Thermal images of the patient were also recorded (Thermovision, A325, Flir Systems, Sweden) at the start and end of each phase of exercise or rest and then again at the end of the cooling intervention. 

Where possible, mean data are provided and standard deviations are shown in parentheses.

## 4. Results

Participant 1 exercised for 15 and 30 min, respectively, for the first and second exercise bouts. Heat production during these exercise bouts was calculated as 143 Watts.m^−2^. The first exercise session was halted on the request of the participant due to feelings of fatigue. The exercise bouts were separated by a passive recovery period (15 min). Following the recovery period, he was able to exercise at the same intensity for 30 min. The rates of the rise in T_ac_ during the first and second intermittent exercise bouts were 2.8 and 1.5 °C.hr^−1^, respectively (Figure 1A). The first recovery session was passive and resulted in the T_ac_ cooling at a rate of 5.6 °C.hr^−1^. For the second active cooling intervention (hand and foot immersion in cold water (temperature of 12.5 °C)), participant 1 was already cooling quickly by the time the hands and feet were immersed. Therefore, the cooling rate using the hand immersion technique was not calculated.

The mean finger skin blood flow at baseline was 351 (56) LDU and an average of 406 (28) LDU during exercise (Figure 1B). Similarly, forearm skin blood flow rose from 97 (47) LDU at the start of exercise to 197 (19) LDU and 252 (40) LDU by the end of the second bout of exercise. 

Whole body sweating was calculated as 0.4 L.hr^−1^ when accounting for fluid consumption and urine output. Local chest and back sweat rates were measured at 0.19 (0.07) and 0.25 (0.1) L.m^2^.hr^−1^, respectively (Figure 1C). In combination with high skin blood flow, skin temperatures were also recorded (mean skin temperature = 29.4 °C), and visible signs of sweating were evident on the infrared images on the skin and clothing (orange patches in the axilla) Figure 2).

Participant 2 exercised for 30 and 20 min during the first and second exercise bouts and passively rested for 20 min in between the two exercise bouts. The second exercise bout was terminated early due to reaching the laboratory withdrawal criterial for elevated T_ac_. T_ac_ rose at rates of 3.4 and 3.1 °C.hr^−1^ for the first and second exercise bouts (Figure 3A).The first recovery session was passive and resulted in T_ac_ cooling at a rate of 0.5 °C.hr^−1^. For the active cooling intervention (hand and foot immersion (water temperature of 12 °C)), the reduction in T_ac_ was at 2.2 °C.hr^−1^. 

In combination with a high skin temperature, participant 2 had a finger skin blood flow value at baseline of 541 (41) LDU and during exercise this was 520 (54 LDU) (Figure 3B). Forearm skin blood flow rose from 41 (19) LDU to 146 (46) LDU during exercise and heat exposure. 

The whole body sweat rate for participant 2 had to be estimated as he poured some of his water bottle over his head. Accounting for this loss, his estimated whole body sweat rate was 0.02 L.hr^−1^. Mean local chest and back sweat rates were 0.03 (0.02) and 0.03 (0.01) L.m^2^.hr^−1^ (Figure 3C). The only signs of sweating were on the palms of his hands and soles of the feet (Figure 4). The remainder of his skin remained hot and dry throughout the heat exposure, with thermal images indicating skin temperatures as high as 38.5 °C at the end of the second exercise bout.

## 5. Discussion

The present case reports demonstrate that control of T_ac_ during exposure to heat and exercise varies in patients affected with ED. It had previously been assumed that all ED patients have reduced sweat production and could not regulate T_ac_ [5]; however, this is not always the case. Sweat production and control of T_ac_ were evident in participant 1 in this case report. Conversely, participant 2 was well-trained, had high skin temperatures from the baseline, high heat production and rapid rate of rise in T_ac_, and no visible sweating (and, in fact, negligible sweat production as indicated from the local sweat rates are within the error of measurement of the equipment), making cooling strategies before, during, and after heat and exercise exposure key to maintenance of T_ac_ and health.

Evidence of sweat production, vasodilation, and control of T_ac_ during exercise were found in participant 1. In fact, during the second recovery session, his sweating responses cooled his T_ac_ to near baseline values prior to placing his hands in cold water, indicating that evaporative cooling was supporting heat loss and use of a fan would augment this through forced convection. Once his hands were in cold water, a vasoconstrictor response in the finger and forearm were immediately observed, illustrating that his T_ac_ was normothermic, and vasoconstriction occurred to preserve his normal body temperature. The combination of this information is indicative of functioning thermoregulation; whether this is adequate during more intense exercise or greater ambient heat exposures could not be determined. However, he was also quick to fatigue when exercising at moderate workloads, a likely consequence of the lack of activity he undertook. Consequently, he was encouraged to undertake more physical activity for weight management and to improve aerobic fitness and muscular endurance. This process of regularly increasing T_ac_ through repeated bouts of exercise also results in partial heat acclimation, which is characterized by a lowering of the sweating threshold [12] and greater sweating sensitivity [13], leading to increased sweating rates [13,14]. Consequently, this provides opportunities for participant 1 to regulate temperature more effectively. It is only with increasing size, mass, and maturation does sweating become more prominent. In fact, sweat rates in men are greater than those of boys [15]. Therefore, the sweat rate of participant 1 may increase as he continues to grow and mature. He will be followed up to establish how his thermoregulatory responses change as he matures, and increases his physical activity level and exposure to elevated T_ac_.

The sweat function in healthy populations varies across the skin surface [10,16]. This can be exacerbated in patients affected with ED where sweating can be patchy [17], and this was observed in participant 2, where sweating was only evident on the palms of his hands and soles of his feet (Figure 4). Active vasodilatation was observed during the exposure to heat and exercise when T_ac_ was elevated, enabling delivery of heat to the skin for convective or conductive heat loss to occur. However, T_ac_ did not plateau during exercise, which is evidence of insufficient thermoregulatory activity. The rapid increase in T_ac_ and an absence of cooling without intervention makes the risk of hyperthermia and heat illness during exercise or heat exposure a likely outcome. At best, this reduces patients’ activities of daily living and at worst could be terminal [4], thus cooling interventions are required to enable exposure to warm ambient environments and participation in exercise.

Provision of effective cooling strategies for participant 2 are required to support thermoregulation, prevent hyperthermia or heat illness, and enable the activities of daily living. The cooling strategy suggested for participant 1 (whole body fanning) would not be a substantial benefit for participant 2, unless his body and clothes were liberally sprayed with water to substitute sweat prior to fanning when it would likely then be very beneficial. His high skin temperature and high peripheral skin blood flow even at baseline indicates he was vasodilated and able to liberate heat from the skin through convective or conductive means. In fact, this combination of elevated T_ac_ and peripheral vasodilation showed that immersion of the hands and/or feet in cold water effectively removed excess heat. Maintenance of peripheral vasodilatation, due to an elevated Tac, enabled the delivery of heat to the skin via the circulation. In fact, these techniques are used to reduce heat strain in healthy populations exposed to hot environments [18] when sweat evaporation is prevented by encapsulating the body in water vapor-impermeable clothing [19]. These cooling strategies are pragmatic, can be customized, and work with the function of the body.

Many cooling strategies rely upon good skin blood flow [20,21,22]. Brengleman et al. [23] studied a family affected with ED, showing that active vasodilatation was absent in response to elevations in T_ac_. Removal of excess heat requires warm blood to perfuse the cutaneous circulation and be cooled. For each liter of blood at 37 °C that flows through the skin and returns to the deeper tissues at 36 °C, the body loses roughly 1 kcal of heat. As the maximum skin blood flow can be as much as 3 to 4 L/min, the same 1 °C fall in the temperature of the blood would offload up to 4 kcal/min (280 W) [20]. Consequently, to remove excess stored heat efficiently, methods of cooling using hand and foot immersion rely upon high skin blood flows to areas in contact with the cooled skin [24,25]. Although the use of an ice vest is more efficient and cools the deep body more quickly if there is a high skin blood flow, if a high skin blood flow is not present, conductive cooling through the layers of tissues still occurs but cools the T_ac_ more slowly. Vasodilatation in the forearm and high finger skin blood flows can be seen in both participants in this case report and therefore immersion of the hands in cold water would be a useful cooling intervention. The differences in the magnitude of the skin blood flow values reported between participants is likely to be due to the site-specific nature of the laser Doppler measurements. In addition, the skin blood flow on the chest was not directly measured, but both participants had good capillary refill when the skin was blanched before and after exercise and heat exposure. Therefore, active cooling using water immersion of the hands and feet or wearing an ice vest would be effective in cooling both participants described here.

Both participants in the present case series had diagnoses of ED, although the mutated gene had not been confirmed in either participant. Identification of the mutated gene is important as key genotype–phenotype associations between skin type and hair type have been made recently [26]. Correctly identifying the gene and consequently the phenotype characteristics (including sweat function) would enable a more tailored message to be conveyed to families affected with ED. This would replace the blanket assumption that all ED patients do not have functioning sweat glands and as a consequence cannot regulate their deep body temperature. Accordingly, this would reduce the likelihood that limits will be placed on physical activity due to fears of uncontrollable hyperthermia (as was the case with participant 1). 

The temperature regulation test used in this case series can be used to establish if vasomotor and sudomotor activity are sufficient to control deep body temperature during exercise and heat exposure. This is also key in those patients who show evidence of sweating, but it is not clear that they sweat sufficiently to regulate deep body temperature. The test can also provide information that would be of utility when heat and exercise exposure is planned. For instance, providing information on the intensity and duration of work that can be completed safely with only a mildly elevated deep body temperature; establish suitable work–rest schedules during heat exposure to ensure that the required ‘work’ is completed whilst giving ample breaks to allow cooling before the next bout of exercise; examine the vasomotor and sudomotor responses to an elevated deep body temperature; and study the effect different cooling interventions have to establish which would be optimal for that individual.

## 6. Conclusions

Assessment of thermoregulation could be routinely performed by clinical physiologists who have training in thermal physiology, and performed in hospitals or laboratories. The equipment and facilities used need not be as complex or expensive as that used in the present case series. They would require access to appropriate exercise equipment and or a warm environment, basic temperature-measuring instruments (skin thermistors, aural, rectal or gastric temperature-measuring devices), scales, and a flow ventilated sweat capsule or use of standard sweat collection capsules. Patients could be referred to these facilities when clinicians find no evidence of sweating and when sweating is evident but may be limited and temperature regulation questioned.

## Figures and Tables

**Figure 1 ijerph-16-04514-f001:**
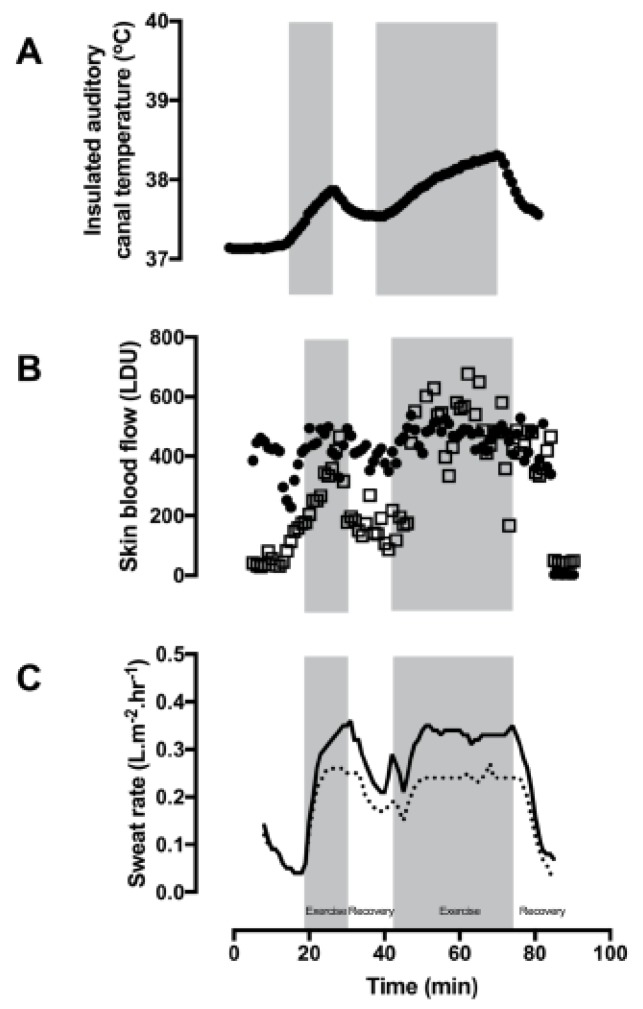
Participant 1 during exercise and recovery in 30 °C air and 50% relative humidity (**A**) Insulated auditory canal temperature (T_ac_), (**B**) finger skin blood flow (closed circles), forearm skin blood flow (open squares), and (**C**) sweat rate on the chest (dashed line) and back (solid line). Exercise was performed during the gray shaded sections.

**Figure 2 ijerph-16-04514-f002:**
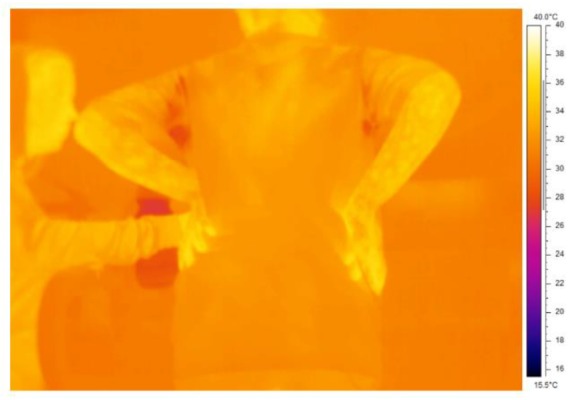
Infrared image of participant 1 at the end of the second exercise phase. Evidence of sweating is apparent on the image as dark orange staining on the t-shirt in contact with the axilla.

**Figure 3 ijerph-16-04514-f003:**
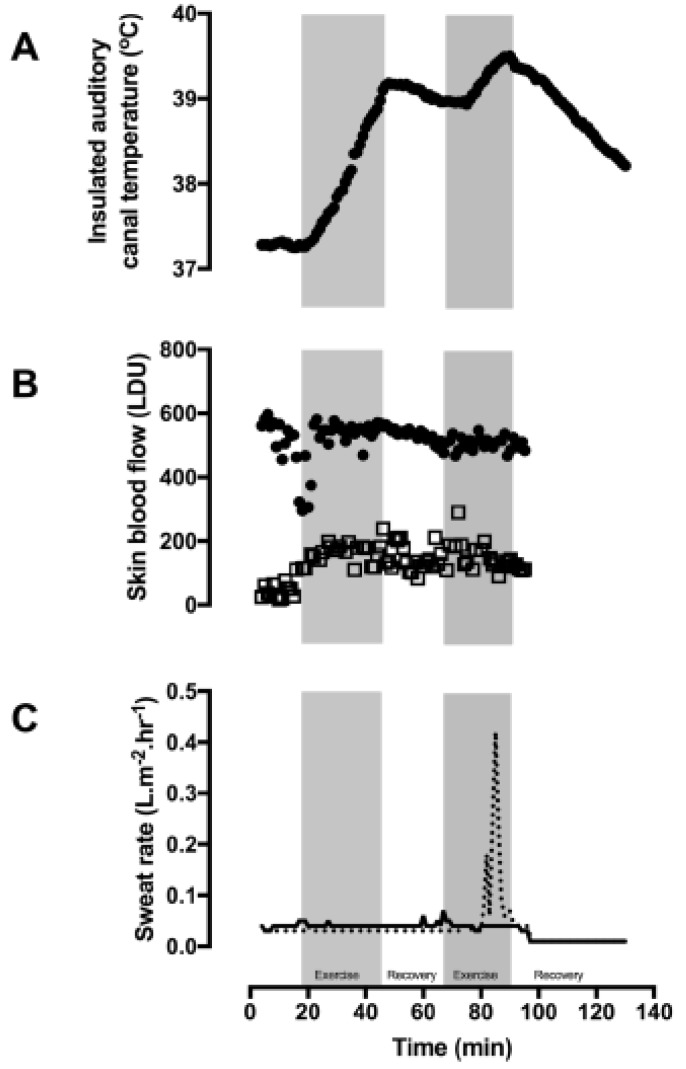
Participant 2 during exercise and recovery in 30 °C air and 50% relative humidity. (**A**) Insulated auditory canal temperature (T_ac_), (**B**) finger skin blood flow (closed circles), forearm skin blood flow (open squares), and (**C**) sweat rate on the chest (dashed line) and back (solid line). Exercise was performed during the gray shaded sections.

**Figure 4 ijerph-16-04514-f004:**
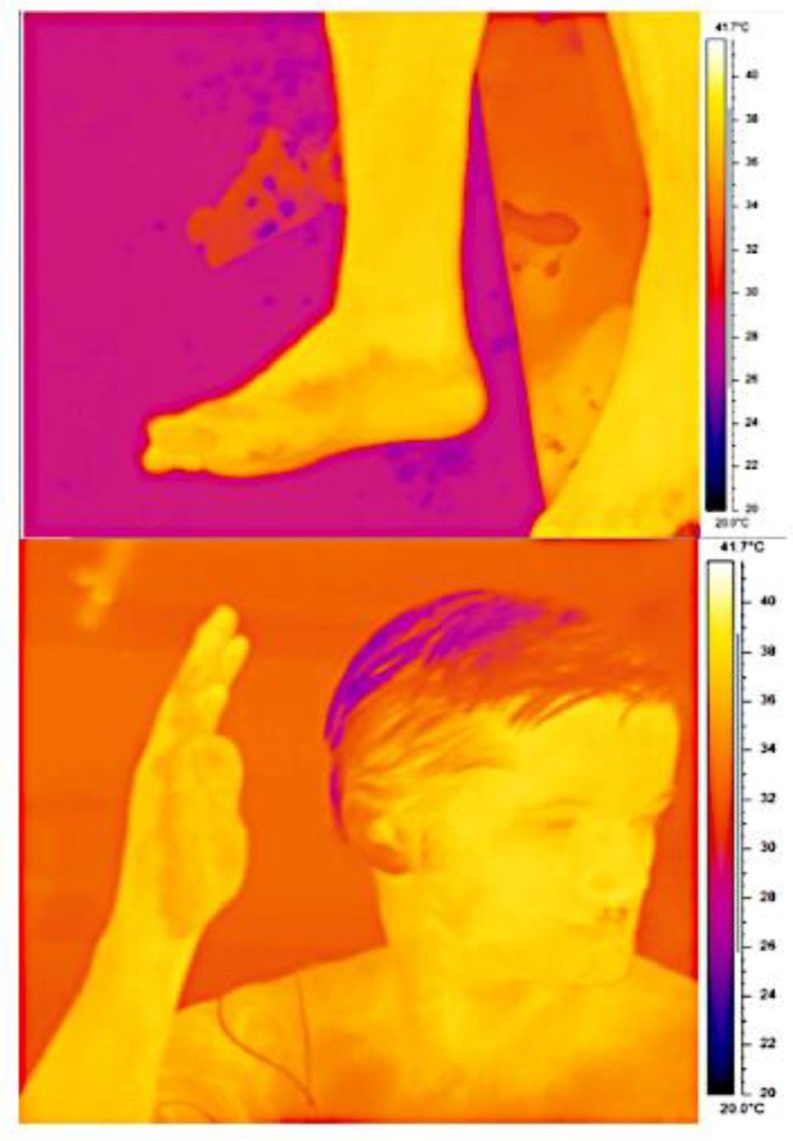
Infrared images of participant 2 at the end of the second exercise phase. The orange coloration of the palm of the hands and soles of the feet indicate sweating.

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
