# Peer review of "Thermoregulation in Ectodermal Dysplasia: A Case Series"

_ijerph, 2019, doi:10.3390/ijerph16224514_

Round 1

Reviewer 1 Report

Overall comments

A really nice case study shedding light onto a poorly understood issue for patients with ED.

 Major comments

The introduction could be improved somewhat. I find the link to the cooling intervention a little tenuous and I think it would be really helpful to highlight the lack of research conducted in this area. Are we aware of any coping mechanisms employed by patients with ED when exercising, or do they simply refrain from exercising? Highlighting a lack of information may also strengthen the rationale for presenting this case study. Furthermore, as a thermophysiologist it makes sense to me to consider cooling interventions as a preventative measures and/or coping strategy. However, I think the readers of this are more likely to come from a clinical/ public health or even applied setting so I think the introduction could be strengthened by explaining some basic thermoregulatory details and also the way in which cooling strategies work (See specific comments in the minor comments section).   

This is a case study of 2 participants, one of which clearly has some thermoregulatory capabilities and the other has almost none. These examples are used to justify providing individual specific cooling strategies but I wonder if you have just recruited two extreme examples? How common is it for someone with ED to be able to sweat? What I am trying to infer is whether you have selected these patients to suit your report? Is it more common to have not sweating capability or is P1 a rare exception? I do not have the experience in this but it would be good to recognize this in your paper?

 Minor comments

Abstract, line 14: should it be ‘could not regulate…’ rather than ‘did not regulate…’? I feel that ‘did not’ suggests a conscious effort whilst ‘could not’ reflects an inability to do something and as sweating is an autonomic response then this may be a better choice of word?

Introduction, line 33-34: Somewhere in the introduction it might be useful to briefly explain thermoregulation, especially the mechanism of heat loss via evaporation and how important it is for humans. As a thermal physiologist I can appreciate this but the potential readers of this journal may not be fully aware and a few sentences may be informative. 

Introduction, line 39: please expand on how they were matched (e.g age, sex, body mass).

Introduction, lines 33-45:  As with my previous comment it might be useful to give a bit more background on cooling interventions, including how they work and some evidence for their use. This would be useful around line 43.

Introduction, line 40: could you please add some descriptive data here about how much higher their ‘deep body temperature’ was? It would be useful here to state whether their temperatures were considered hyperthermic or just higher than their matched controls.

Participant 1 and 2: you state that P1 has a clinical diagnosis and P2 has a genetic diagnosis. I am unfamiliar with the difference. If this is important then it may be useful to state this somewhere in the report.

Materials and Methods:

I find this section very short and lacking more detailed information. I suggest adding more information particularly to the part where you describe the equipment but do not necessarily state what they are used for. There is also no rationale for the locations used for the sweat capsule, laser doppler, skin heat flux sensors etc.

Materials and Methods, Line 74: You mention that thermal imaging was takena t the end of each phase but at this point it is not clear what this refers to. The order seems a little jumbled.

Materials and Methods, Line 78: do you mean ‘intention’?

Materials and Methods, Line 79: I think the abbreviation has not previously been explained.

Materials and Methods, Line 80: why do you state ‘ideally’?

Materials and Methods, Line 82: why did you select 10-12C for the cooling? And how long was the cooling for?

Results. As this is a case study it would be good to expand why the participants could not complete the different sections of the protocol.

Results: Have you standarised the skin blood flow measure (e.g. change from baseline or relative to a maximum value)? I think this is standard practice when reporting SkBf, is it not?

Results, line 100: It is not really clear from the image that sweating was evident. Can you change the colour range on the image to highlight where the sweating occurred? If this is not possible then describe how sweating can be seen from this thermal image.

Results: throughout the results and in this paper, you refer to ‘deep body temperature’ but you measure this using the auditory canal temperature. I think this is misleading especially for the readers of this journal who may not be aware of the issues with this measurement, especially during an exercising protocol. It would be better to refer to what it actually is (i.e. Tau) throughout the document for clarity.

Results, line 113-115: You state the baseline finger skin blood flow and then state the mean skin blood flow during exercise. Is this correct or are they both finger skin blood flow? If the latter then this seems quite interesting. Could you explain why they have such a high resting finger skin blood flow response in the discussion? If the former, then is this the mean skin blood flow a mean of the finger and the forearm? This is not clear to me.

Discussion, line 141-145: I am aware that this is a case study but the writing style seems very anecdotal here. I find the section about the patient fatiguing too quickly and the advice given to exercise and manage his weight etc. to be uninformative to the case report. This is a personal issue for the patient which adds nothing to the report, I recommend you delete this sentence.

Discussion, line 145-148: Whilst this sentence is correct, it seems out of place or at least not explained in context. Could you elaborate on this or explain why I would be important for this particular patient.

Discussion: Participant 2 is an interesting case study and it would be great to use this opportunity to really explain the risks for the patient. Could you elaborate more on the lack of his physiological responses and what the implications are. This would be followed by your elaborate discussion of potential coping strategies (lines 169 onwards)

Discussion, line 158-160: ‘maintenance of health’ …I am not convinced by this sentence. It would be ideal to explain this in relation to thermoregulation. Perhaps expand on this?

Author Response

Dear Professor Mundel,

May I thank you and the reviewers for taking the time to consider this manuscript and for providing timely and insightful comments. We are grateful for the kind words given from the reviewer. We have attempted to address their comments individually below. We feel these edits have substantially strengthened the manuscript. Thank you      

For clarity our responses are written in bold. All line numbers referenced refer to the clean version of the manuscript

Reviewer 1

Overall comments

A really nice case study shedding light onto a poorly understood issue for patients with ED.

Major comments

The introduction could be improved somewhat. I find the link to the cooling intervention a little tenuous and I think it would be really helpful to highlight the lack of research conducted in this area. Are we aware of any coping mechanisms employed by patients with ED when exercising, or do they simply refrain from exercising? Highlighting a lack of information may also strengthen the rationale for presenting this case study. Furthermore, as a thermophysiologist it makes sense to me to consider cooling interventions as a preventative measures and/or coping strategy. However, I think the readers of this are more likely to  come from a clinical/ public health or even applied setting so I think the introduction could be strengthened by explaining some basic thermoregulatory details and also the way in which cooling strategies work (See specific comments in the minor comments section). 

Our focus in performing the assessments is to establish if sweating is present and therefore a capability to thermoregulate. Therefore, post exercise cooling was the main focus of the cooling intervention at this stage. However, as you correctly point out. Patients are advised to try to limit heat gain during exercise by using cooling interventions.

This is a case study of 2 participants, one of which clearly has some thermoregulatory capabilities and the other has almost none. These examples are used to justify providing individual specific cooling strategies but I wonder if you have just recruited two extreme examples? How common is it for someone with ED to be able to sweat? What I am trying to infer is whether you have selected these patients to suit your report? Is it more common to have not sweating capability or is Pl a rare exception? I do not have the experience in this but it would be good to recognize this in your paper? Thank you for your comments, this is a very valid point.

We chose two patients from the cohort of patients we have collected data from whom had good quality data and were prepared to consent for their anonymised data to be used. We have data from others patients with similar presentations of both no sweat and evidence of sweating, as well as patients who have some moderate sweating capability. Typically, it was initially believed that patients with Ectodermal Dysplasia could not sweat, however through discussion with patients and the patient support group the amount and patterning of sweating varies. Some having the appearance of sweating, similar to that of unaffected siblings, whilst others have no sweat function. Therefore, the two cases were chosen as being representative of the group that we have monitored at this stage.

Minor comments

Abstract, line 14: should it be 'could not regulate...' rather than 'did not regulate...'? I feel that 'did not' suggests a conscious effort whilst 'could not' reflects an inability to do something and as sweating is an autonomic response then this may be a better choice of word?

This has been amended

Introduction, line 33-34: Somewhere in the introduction it might be useful to briefly explain thermoregulation, especially the mechanism of heat loss via evaporation and how important it is for humans. As a thermal physiologist I can appreciate this but the potential readers of this journal may not be fully aware and a few sentences may be informative.

Introduction, line 39: please expand on how they were matched (e.g age, sex, body mass).

Introduction, lines 33-45: As with my previous comment it might be useful to give a bit more background on cooling interventions, including how they work and some evidence for their use. This would be useful around line 43.

Introduction, line 40: could you please add some descriptive data here about how much higher their 'deep body temperature' was? It would be useful here to state whether their temperatures were considered hyperthermic or just higher than their matched controls.

The responses to the four comments have been incorporated as one and amended as follows:

Many patients affected with ED experience difficulties regulating their temperature and can develop hyperthermia as a result of reduced or absent sweating. Sweat production on the surface of the skin aids heat loss through evaporation. Consequently, a reduction or the absence of sweating may result in excess heat storage and the development of hyperthermia (a deep body temperature greater than 39°C) and heat illness (including heat exhaustion or heat stroke) or death. [4] Unpublished data from the UK patient group of the Ectodermal Dysplasia Society (EDS), suggests that deep body temperature measurements of affected children are hotter than siblings without ED, when measured three times a day for a week. In that study, temperatures were recorded using infrared ear thermometers three times a day, for one week. More recently, a controlled laboratory-based study was performed [6] by exercising children with ED and matched controls (matched for sex and age) in 25 °C air temperatures (40% relative humidity). The authors report that the children with ED had deep body temperatures 1 °C higher after a maximum of 30 minutes of cycling exercise at a moderate to high intensity. Ninety minutes after the exercise deep body temperatures of the ED group were still elevated above their starting temperatures where the control had cooled with in 30 minutes of stopping. This suggests that children with ED have a rapid and possibly dangerous rise in temperature and may take longer to cool. In those affected by ED, cooling interventions could help to reduce the thermal stain experienced and increase the rate of cooling in recovery from exercise or heat exposure. Cooling interventions can increase heat loss through conductive, convective or evaporative means. The effectiveness of the intervention may be dependent on the location of the cooling and if a viable peripheral circulation can be maintained.[7,8] For instance, post exercise ice vest use in ED patients resulted in greater rates of cooling in comparison to exercise recovery without cooling.[4] However, research in healthy volunteers exposed to high ambient temperatures showed whole body spraying and fanning and hand immersion in cold water had greater cooling power than air, liquid or phase change cooling vests [7].

Participant 1 and 2: you state that Pl has a clinical diagnosis and P2 has a genetic diagnosis. I am unfamiliar with the difference. If this is important then it may be useful to state this somewhere in the report.

Further information has been added to the introduction and to the case history:   

Introduction: 

The condition can be diagnosed either genetically (from a blood test or mouth swab) if there is a known family history, or clinically by the appearance of a triad of disorders: few and often pointed teeth (hypodontia), fine and sparse hair (hypotrichosis) and diminished or absent sweat function (hypohidrosis), although there are many other conditions (including cleft lip and palate, eczema, xerophthalmia, xerostomia) which can co-exist and variably affect the patient. [1]

Case history:

He has a clinical diagnosis of ED, based on the presentation of hypodontia, hypotrichosis and assumption of hypohidrosis.

Materials and Methods:

I find this section very short and lacking more detailed information. I suggest adding more information particularly to the part where you describe the equipment but do not necessarily state

what they are used for. There is also no rationale for the locations used for the sweat capsule, laser Doppler, skin  heat  flux sensors etc.     

Information has been added to this section, between lines 77 to 110.

Materials and Methods, Line 74: You mention that thermal imaging was  taken at the end of  each phase but at this point it is not clear what this refers to. The order seems a little jumbled.

This information has been added

Thermal images of the patient were also recorded (Thermovision, A325, Flir Systems, Sweden) at the start and end of each phase of exercise or rest and then again at the end of the cooling intervention.

Materials and Methods, Line 78: do you mean 'intention'?

This has been amended

Materials and Methods, Line 79: I think the abbreviation has not previously been explained.

Tac been included on first use

Materials and Methods, Line 80: why do you state 'ideally'?

This has been removed

Materials and Methods, Line 82: why did you select 10-12C for the cooling? And how long was the cooling for

This water temperature was readily available from the water supply and has also been used in previous research, the following was added to the manuscript

This temperature was readily available from the tap and was used in previous research. [8]

Results. As this is a case study it would be good to expand why the participants could not complete the different sections of the protocol.

This information has been added as following for participant 1:

The first exercise session was halted on the request of the participant, due to feelings of fatigue. The exercise bouts were separated by a passive recovery  period (15 minutes). Following the recovery period, he was able to exercise at the same intensity for 30 minutes.

And for participant 2:

The second exercise bout was terminated early due to reaching the Laboratory withdrawal criteria! for elevated Tac.

Results: Have you standarised the skin blood flow measure (e.g. change from baseline or relative to a maximum value)? I think this is standard practice when reporting SkBf, is it not?

This information has been added to the methods section

'Laser Doppler sensors placed on the middle finger pad and ventral sur face of the forearm (Moor Instruments, UK) were used to measure skin blood flow, the Laser Doppler probes were calibrated using a microsphere solution (Moors instruments, UK) and biological zero assessed by occluding flow to the forearm or digit. Data are corrected for biological zero.

Results, line 100: It is not really clear from the image that sweating was evident. Can you change the colour range on the image to highlight where the sweating occurred? If this is not possible then describe how sweating can be seen from this thermal image.

It was not possible to change the range without losing definition within the picture, therefore a better explanation was included in the figure legend. Please see below

Figure 2. Infrared image of participant 1 at the end of the second exercise phase. Evidence of sweating is apparent on the image as dark orange staining on the t-shirt in contact with the axilla.

Results: throughout the results and in this paper, you refer to 'deep body temperature' but you measure this using the auditory canal temperature. I think this is misleading especially for the readers of this journal who may not be aware of the issues with this measurement, especially during an exercising protocol. It would be better to refer to what it actually is (i.e. Tau) throughout the document or clarity.

This has been amended throughout where we refer to our results

Results, line 113-115: You state the baseline finger skin blood flow and then state the mean skin blood flow during exercise. Is this correct or are they both finger skin blood flow? If the latter then this seems quite interesting. Could you explain why they have such a high resting finger skin blood flow response in the discussion? If the former, then is this the mean skin blood flow a mean of the finger and the forearm? This is not clear to me.

This information has been amended in the results section to better explain the finger skin blood flow data.

In combination with high skin temperature Participant 2 had a finger skin blood flow value at baseline of 541 (41) LDU and during exercise 520 (54 LDU) (Figure 3B).

And the following added to the discussion:

Sweat function in healthy populations varies across the skin surface, [10,17] this can be exacerbated in patients affected with ED where sweating can be patchy,  [18] this  was  observed  in Participant  2, where sweating was only evident on the palms of his hands and soles of his feet (Figure 4). Active vasodilatation was observed during  the exposure  to heat  and  exercise  when Tac was  elevated, however Tac did not plateau during exercise, as such there was no evidence of thermoregulation.

Provision of appropriate cooling strategies for Participant 2 are required to support thermoregulation, prevent hyperthermia or heat illness and enable more normal activities of daily living. The cooling strategy suggested for participant 1 (whole  body  fanning)  would  not  be  a substantial benefit  for  Participant  2, unless he and his  clothes  were liberally  sprayed  with  water  to substitute sweat prior  to  fanning, when it would then likely be very beneficial. His high skin  temperature  and  high  peripheral  skin  blood  flow even at  baseline  indicate  he  was  vasodilated  to  liberate  heat  from  the  skin  through  convective  or conductive means. In fact, this combination of elevated Tac and peripheral  vasodilation  showed  that immersion of the hands and/or feet in cold water effectively removed  excess heat.  Maintenance of peripheral vasodilatation,  due to an  elevated Tac,  enabled  delivery  of  heat to  the  skin  through the circulation. In fact, these techniques are used  to  reduce  heat  strain  in  healthy  populations  exposed  to  hot environments [19] when sweat evaporation is reduced by  encapsulating  the  body  in  water vapour impermeable clothing.[20] These cooling strategies are pragmatic, can be customized, and  work  with  the function of the body.

Discussion, line 141-145: I am aware that this is a case study but the writing style seems very anecdotal here. I find the section about the patient fatiguing too quickly and the advice given to exercise and manage his weight etc. to be uninformative to the case report. This is a personal issue for the patient which adds nothing to the report, I recommend you delete this sentence.

We agree that the writing style is quite anecdotal, but we feel that it is important to recognise that part of problem participant 1 faced have been caused by the lack of physical activity and exposure to exercise, because of the parents’ rigid adherence to medical advice founded on an assumption of lack of sweat production rather than evidence of the level of sweat production, which this testing has now made available to the family. Therefore, the language has been tightened up, but the message has been kept in the paper, and included as follows:

Once his hands were in cold water a vasoconstrictor response in the finger and forearm were immediately observed, illustrating that his Tac was normothermic, and vasoconstriction occurred to preserve normal body temperature. The combination of this information is indicative of functioning thermoregulation; whether this is adequate during more intense exercise or greater ambient temperatures, could not be determined. However, he was also quick to fatigue when exercising at moderate workloads; a likely consequence of the lack of activity he has been allowed to undertake previously. Consequently, he was encouraged to undertake more physical activity for weight management, to improve aerobic fitness and muscular endurance.

Discussion, line 145-148: Whilst this sentence is correct, it seems out of place or at least not explained in context. Could you elaborate on this or explain why I would be important for this particular patient.

This information has been added:

This process of regularly increasing Tac through repeated bouts of exercise also results in partial heat acclimation which is characterized by a lowering of the sweating threshold [13] and greater sweating sensitivity [14] leading to increased sweating rates [14,15]. Consequently, providing opportunities for participant l to regulate temperature more effectively.

Discussion: Participant 2 is an interesting case study and it would be great to use this opportunity to really explain the risk s for the patient. Could you elaborate more on the lack of his physiological responses and what the implications are. This would be followed by your elaborate discussion of potential coping strategies (lines 169 onwards)

This information has been amended as below:

Sweat function in healthy populations  varies across the skin surface [10,17]; this can be exacerbated in patients affected with ED where sweating can be patchy [18] and this was observed in Participant 2, where sweating was only evident on the palms of his hands and soles of his feet (Figure 4). Active vasodilatation was observed during the exposure to heat and exercise when Tac was elevated, enabling delivery of heat to the skin for convective or conductive heat loss to occur. However Tac, did not plateau during exercise, as such the evidence is of insufficient thermoregulatory activity. The rapid increase in Tac, an absence of cooling without intervention, makes the risk of hyperthermia and heat illness during exercise or heat exposure a likely outcome. At best this reduces the patient’s activities of daily living and at worst could be terminal [4}, thus cooling interventions are required to enable exposure to warm ambient environments and participation in exercise.

Discussion, line 158-160: ' maintenance of health' ...I am not convinced by this sentence. It would be ideal to explain this in relation to thermoregulation . Perhaps expand on this?

This has been changed to:

Provision of appropriate cooling strategies for Participant 2 are required to support thermoregulation, prevent hyperthermia or heat illness and enable activities of daily living.

Reviewer 2 Report

It‘s very interesting to study thermoregulation in ectodermal dysplasia. Many gene mutaions can lead to ED.  If authors can find gene mutations in two participants, we can better understand the relationship between genotype and phenotype.

Author Response

Dear Professor Mundel,

May I thank you and the reviewers for taking the time to consider this manuscript and for providing timely and insightful comments. We are grateful for the kind words given from the reviewer. We have attempted to address their comments individually below. We feel these edits have substantially strengthened the manuscript. Thank you      

For clarity, our responses are written in bold. 

Reviewer 2

It's very interesting to study thermoregulation in ectodermal dysplasia. Many gene mutations can lead to ED. If authors can find gene mutations in two participants, we can better understand the relationship between genotype and phenotype.

Thank you for your comments.